# Improvement of Biophysical Properties and Affinity of a Human Anti-L1CAM Therapeutic Antibody through Antibody Engineering Based on Computational Methods

**DOI:** 10.3390/ijms22136696

**Published:** 2021-06-22

**Authors:** Heesu Chae, Seulki Cho, Munsik Jeong, Kiyoung Kwon, Dongwook Choi, Jaeyoung Lee, Woosuk Nam, Jisu Hong, Jiwoo Lee, Seonjoo Yoon, Hyojeong Hong

**Affiliations:** 1Department of Systems Immunology, Kangwon National University, Chuncheon 24341, Korea; chs@apitbio.com (H.C.); anstlrj@kangwon.ac.kr (M.J.); gykwon@kangwon.ac.kr (K.K.); ghdwltn55@kangwon.ac.kr (J.H.); snm04062@kangwon.ac.kr (J.L.); 2APIT BIO Inc., B910, Munjeongdong Tera Tower, 167 Songpa-daero, Songpa-gu, Seoul 05855, Korea; liy5999@apitbio.com (J.L.); wsnam@apitbio.com (W.N.); 3Institute of Bioscience and Biotechnology, Kangwon National University, Chuncheon 24341, Korea; seul8502@naver.com; 4Division of Drug Process Development, New Drug Development Center, Osong Medical Innovation Foundation, Chungcheongbuk-do, Cheongju-si 28160, Korea; dwchoi@kbiohealth.kr

**Keywords:** therapeutic antibody, anti-cancer antibody, antibody engineering, biophysical properties, computational methods, research cell bank

## Abstract

The biophysical properties of therapeutic antibodies influence their manufacturability, efficacy, and safety. To develop an anti-cancer antibody, we previously generated a human monoclonal antibody (Ab417) that specifically binds to L1 cell adhesion molecule with a high affinity, and we validated its anti-tumor activity and mechanism of action in human cholangiocarcinoma xenograft models. In the present study, we aimed to improve the biophysical properties of Ab417. We designed 20 variants of Ab417 with reduced aggregation propensity, less potential post-translational modification (PTM) motifs, and the lowest predicted immunogenicity using computational methods. Next, we constructed these variants to analyze their expression levels and antigen-binding activities. One variant (Ab612)—which contains six substitutions for reduced surface hydrophobicity, removal of PTM, and change to the germline residue—exhibited an increased expression level and antigen-binding activity compared to Ab417. In further studies, compared to Ab417, Ab612 showed improved biophysical properties, including reduced aggregation propensity, increased stability, higher purification yield, lower pI, higher affinity, and greater in vivo anti-tumor efficacy. Additionally, we generated a highly productive and stable research cell bank (RCB) and scaled up the production process to 50 L, yielding 6.6 g/L of Ab612. The RCB will be used for preclinical development of Ab612.

## 1. Introduction

Therapeutic monoclonal antibodies (mAb) are the leading class of drugs on the biopharmaceutical market, largely due to their high specificity, affinity, potency, and their long in vivo half-life. Since approval of the first mAb (OKT3) in 1983, 79 therapeutic antibodies have been approved by FDA and are currently on the market, including 30 mAbs for cancer treatment [1]. Innovations in antibody engineering technologies—such as humanization of murine mAbs, phage display and transgenic mice for generating fully human mAbs, Fc engineering, antibody-drug conjugates, and bispecific antibodies—have contributed to the development of these important drugs [2,3,4,5,6,7,8,9]. Therapeutic antibody discovery and development starts with early candidates (hits), followed by selection of advanced candidates (leads). A critical element of antibody development is termed chemistry, manufacturing, and controls (CMC), which includes construction of an antibody-producing cell line, development of a manufacturing process, and development of suitable analytical methods to validate the antibody’s safety and efficacy [10,11,12].

During the manufacturing process, including cell culture and downstream processing, therapeutic antibodies are at risk of physical and chemical degradation through multiple pathways [13]. Degradation may affect antigen binding, decrease antibody efficacy, or even lead to immunogenic products [14,15,16]. Protein aggregation is the most common and substantial type of physical degradation associated with therapeutic antibodies [13,17,18]. High concentrations are associated with increased protein–protein interaction frequency, which proportionally increases the opportunity for aggregation formation. Changes in extrinsic conditions—including pH, salt, temperature, shaking, and viscosity—can also promote protein–protein associations that can lead to aggregation events [13,19,20,21,22]. 

Post-translational modifications (PTMs) may also cause problems during therapeutic antibody development. For example, asparagine (Asn) deamidation, the most common pathway for the chemical degradation of therapeutic antibodies, results from hydrolysis of the amide side-chain of Asn, which cumulatively produces a heterogeneous mixture of aspartate (Asp) and isoAsp at the affected position [23,24]. Asn residues are more prone to deamidation when they are in a solvent-accessible region or are followed by a small or flexible residue, such as serine (Ser) or glycine (Gly). Asn deamidation can affect function if it occurs at a binding interface, such as the complementarity determining regions (CDRs) of an antibody molecule [19,23,25]. Another PTM, Asp isomerization, involves the non-enzymatic interconversion of Asp and isoAsp residues [19]. Asp isomerization may occur more commonly when Asp is followed by Ser, Gly, or Asp; it can affect protein function when it occurs in CDRs; and it can potentially result in fragmentation [26,27].

Each mAb has unique biophysical properties, mainly due to differences in the CDR residues and framework scaffolds. Thus, the identification of degradation-prone or unstable regions early in antibody development could allow for re-engineering of leads. This approach is aided by computational modeling tools that predict regions susceptible to physical and chemical degradation [28,29,30]. To develop a therapeutic antibody with anti-tumor activity, we previously isolated a human mAb (Ab4) that specifically binds to human and rodent L1 cell adhesion molecule (L1CAM) from a human naïve Fab library using phage display [31]. We next generated an affinity-matured version (Ab417) of this hit through site-directed mutagenesis of CDR residues, and we validated its anti-tumor efficacy and mechanism of action in rodent models [31,32].

In the present study, we aimed to improve the biophysical properties of the lead antibody (Ab417). We analyzed the heavy (VH) and light (VL) chain sequences and the three-dimensional (3D) model of Ab417, using computational methods to identify potential PTMs and to calculate aggregation propensities. Next, we designed 20 variants of Ab417 with reduced aggregation propensity, fewer potential PTM motifs, and the lowest predicted immunogenicity. We constructed these Ab417 variants and analyzed their expression levels and antigen-binding activities. One variant (Ab612), which was generated by substituting the four VH residues and two VL residues of Ab417, exhibited a higher expression level and higher antigen-binding activity compared to Ab417. Further studies demonstrated that compared to Ab417, Ab612 also showed higher productivity, improved biophysical properties, and a higher affinity and in vivo anti-tumor efficacy. For the preclinical development of Ab612, we generated a highly productive and stable research cell bank (RCB), and scaled up the production process to 50 L, which yielded 6.6 g/L of the antibody. Ab612 is now considered a candidate antibody to progress to preclinical development.

## 2. Results

### 2.1. Design and Selection of an Ab417 Variant with Improved Biophysical Properties

To design Ab417 variants with reduced aggregation propensity and increased stability, we first aligned the VH and VK sequences of Ab417 with a set of human germline genes and screened for potential PTMs, while constructing a structural model of the Fv of Ab417. Secondly, we calculated the sequence- and structure-based aggregation propensities to identify aggregation hotspots and made substitutions with the aim of reducing the aggregation propensity and improving the stability, using Lonza’s AggreSolve^TM^ in silico tools. We evaluated the aggregation propensity by analyzing regions outside the CDRs with high aggregation propensity, screening for positions with potentially problematic PTMs, identifying positions that were out of line with a conserved consensus in the human germline genes, and screening for solvent-exposed aliphatic or hydrophobic residues. Thirdly, we screened the engineered sequences of Ab417 for Th epitopes using Epibase^TM^ in silico tools to ensure that aggregation-reducing substitutions would not increase the immunogenic potential. Finally, we designed 19 Ab417 variants with engineered VH and VK (Appendix A).

To select the Ab417 variants with improved biophysical properties, the genes coding for the VH and VK of the variants were synthesized and individually subcloned into the heavy and light chain expression plasmids, respectively. Then, the variant combinations were introduced into HEK293F cells for transient expression. After six days of cell cultivation, the culture supernatants were analyzed by quantitative ELISA and indirect ELISA. The results revealed that the variant H3L7 exhibited a higher expression level and retained the same antigen binding activity compared to Ab417. In contrast, the other variants with higher expression levels exhibited lower antigen-binding activities compared to Ab417 (data not shown).

The sequence differences between Ab417 and H3L7 include two substitutions in the VL of H3L7 (I31S and V96S) and four in the VH (R16G, D54E, K76A, and P88A), as shown in Figure 1. The substitutions I31S and V96S in the VL were designed to reduce the surface hydrophobicity in the light chain CDR1 (LCDR1) and LCDR3, respectively. The substitutions in the VH—R16G in the FR1 and K76A in the FR3—were designed to reduce the number of positive charges. Compared to other therapeutic antibodies on the market, Ab417 has a high isoelectric point (pI, 9.6), and large numbers of positive charges may cause the antibody to readily interact with any negatively charged molecules. The D54E substitution was designed to remove two potential PTMs (deamidation and isomerization) in the heavy chain CDR2 (HCDR2) without affecting the binding affinity. The P88A substitution was included because all of the most similar germline genes contain an alanine instead of proline at position 88 of the VH.

To further improve the variant H3L7, we substituted Pro96 for Ser96 in the LCDR3 to reduce the flexibility of the LCDR3 loop. The LCDR3 of Ab417 has a length of 10 residues rather than 9, with no conserved proline residue at position at 95a, and also has two flexible glycine residues. The resulting H3L7 variant (Ab612), as well as Ab417, were transiently expressed in HEK293F cells. Analysis of the culture supernatants by ELISA revealed that the expression levels and antigen-binding activities of Ab612 were higher than those of Ab417 (data not shown). Therefore, Ab612 was selected for further studies.

### 2.2. Expression Analysis and Optimization of Purification Process of Ab417 and Ab612

To precisely compare the expression levels between Ab417 and Ab612, each antibody was expressed using the ExpiCHO-S transient expression system, and the culture supernatant was subjected to quantitative ELISA. The results indicated that the expression level of Ab612 was 2.6-fold higher than that of Ab417 (Figure 2A).

To compare the quality of the produced antibodies, Ab417 and Ab612 were purified from the culture supernatants via three purification steps using Protein A affinity chromatography, cation-exchange chromatography (CIEX), and size exclusion chromatography (SEC) (Figure 2B–D). In the case of Ab417, high-molecular-weight aggregates could be separated from the IgG monomer by stepwise gradient elution in CIEX because they were not separated by linear gradient elution. On the other hand, Ab612 was eluted as a single peak by linear gradient elution (Figure 2C). After SEC, the purification yields of Ab417 and Ab612 were 43% and 61%, respectively, indicating that the purification yield of Ab612 was 40% higher compared to that of Ab417. SEC-HPLC was conducted to validate the purity and homogeneity of the purified antibody. A fraction of fragmented Ab417 was detected, whereas fragmented Ab612 was not detected (Figure 2E).

### 2.3. Thermal Stability of Ab417 and Ab612

To assess the thermal stability of Ab417 and Ab612, the purified antibodies were evaluated by dynamic light scattering (DLS) while increasing the temperature from 30 °C to 80 °C. The aggregation starting point was 74 °C for Ab417 and 77 °C for Ab612 (Figure 3), indicating that Ab612 exhibited increased thermal stability compared to Ab417. Taken together with the analyses of expression and purification processing, these results demonstrate that compared to Ab417, Ab612 showed improved biophysical properties, including increased productivity, reduced aggregation propensity and fragmentation, and increased thermal stability.

### 2.4. Affinity and pI Values of Ab417 and Ab612

To compare the antigen-binding affinities of Ab417 and Ab612, we determined the affinities of the purified antibodies using both competitive ELISA and bio-layer interferometry (BLI) with Octet Red384. We measured the affinities for both human and mouse L1CAM because Ab417 is cross-reactive with rodent L1CAM [31]. In a competitive ELISA, the affinities (KD) of Ab417 and Ab612 for human L1CAM were 0.50 nM and 0.26 nM, respectively (Figure 4A), while both antibodies showed the same affinity for mouse L1CAM (KD, 0.12 nM) (Figure 4B). As determined by BLI, the affinities of Ab417 and Ab612 for human L1CAM were 0.26 nM and 0.11 nM, respectively, while the antibodies showed identical affinities for mouse L1CAM (KD, 0.10 nM) (Figure 4C). These results indicated that Ab612 exhibited a 2-fold higher affinity for human L1CAM compared to Ab417, but not an increased affinity for mouse L1CAM. This demonstrated that the increased affinity of Ab612 was due to an increased intrinsic affinity, but not to its improved biophysical properties.

In the VH of Ab417, two positively charged amino acid residues (Arg16 and Lys76) were substituted with Gly16 and Ala76 to construct Ab612. Thus, we measured the pI of the purified antibodies by capillary isoelectric focusing (cIEF). As expected, the pI of Ab612 (9.25) was lower than that of Ab417 (9.62) (Appendix A).

### 2.5. In Vivo Anti-Tumor Activities of Ab417 and Ab612

Since we confirmed that Ab612 showed improved biophysical properties and affinity compared to Ab417, we next investigated its in vivo anti-tumor efficacy in a cholangiocarcinoma xenograft nude mouse model where Ab417 inhibited tumor cell proliferation.Balb/c nude mice (*n* = 8) bearing Choi-Ck xenografts were i.v. injected with Ab417 or Ab612 (10 mg/kg) or vehicle three times per week for three weeks, and the tumor volume and body weights were measured. At 22 days after the first injection, tumor tissues were removed and weighed. Compared to the control, Ab417 and Ab612 resulted in tumor growth inhibition of 55.5% and 78.2%, respectively, based on mean tumor weight, while not affecting body weight (Figure 5A–D). These results indicated that Ab612 exhibited enhanced anti-tumor efficacy compared to Ab417 in the cholangiocarcinoma model.

### 2.6. Generation of an RCB for Preclinical Development of Ab612

To initiate the preclinical development of Ab612, stable and high-producing Chinese hamster ovary (CHO) cell clones were generated according to the regulatory guidelines. Figure 6A presents the overall scheme for RCB development. Briefly, the heavy and light chain genes were codon-optimized and cloned into the manufacturer’s expression vector, followed by transfection into CHO-K1 host cells. The transfected cells were seeded in 96-well plates (96 WPs) to prepare mini pools. These mini pools were gradually screened from 96 WPs to 24 WPs, cell culture plates (TPPs), and 125 mL shake flasks, based on the titer. Finally, the top 8 mini pools were selected. Next, 117 monoclones were isolated from the 8 mini pools by limiting dilution and were gradually screened from 96 WPs to 24 WPs, TPPs, and shake flasks, according to the titer. After the top 10 clones were amplified for RCB building, the top 8 RCBs were selected based on a fed-batch assay, including antibody purification using Protein A, and quality analysis by UPLC-MS, SEC, and CIEF (data not shown). Finally, the top RCB (006-M71-14), which showed the highest cell line stability, was incubated for 11 days, and then passaged every 3 days until passage 23 (p23). Over this time, the titer and cell-specific productivity (Qp) were measured by PA-HPLC. As shown in Figure 6B, the titers of RCB 006-M71-14 were 3120 (p3), 2577 (p11), 2891 (p18), and 2798 mg/mL (p23), and the Qps were 30.98 (p3), 32.38 (p11), 30.65 (p18), and 29.15 (p23) pg/cell/day. These results indicated that the RCB stably maintained its high productivity during the 80 days of culture.

To assess the quality of the product from RCB 006-M71-14, Ab612 was purified from the culture supernatants by Protein A affinity chromatography and analyzed by SEC-HPLC. The percentages of IgG monomer and high-molecular-weight aggregates of Ab612 were 97.53% and 2.47%, respectively, with no fragmented antibody detected (Figure 6C). Additionally, the purified Ab612 exhibited the same antigen-binding activity as that from transient expression in ExpiCHO-S cells in a sandwich ELISA (Figure 6D). Finally, to test scalability, the fed-batch of the RCB was separately scaled up to 3.7 L and 50 L. After 14 days of cultivation, the titers of 3.7 L and 50 L cultures were determined to be 6.3 and 6.6 g/L, respectively. This demonstrated that the RCB was highly productive and stable, and thus it will be suitable to produce material for preclinical toxicology studies and clinical studies.

## 3. Discussion

The early phase of drug discovery is focused on antibody selection based on specificity, affinity, and functional properties. However, as an antibody is advanced into preclinical development, it is important to consider its biophysical properties to determine whether it can be successfully developed into an efficacious drug. The biophysical properties of a therapeutic antibody can critically influence its late-stage developability, which requires high-level expression, high solubility, conformational and colloidal stability, low poly-specificity, and low immunogenicity [33].

In our previous attempt to develop a therapeutic antibody with anti-tumor activity, we isolated a human mAb that cross-reacts with rodent L1CAM from a human naïve antibody library, and we then generated an affinity-matured version (Ab417) of this hit and validated its anti-tumor efficacy and mechanism of action in rodent models [31,32]. In the present study, we attempted to optimize Ab417 by improving its biophysical properties.

We designed 19 variants with reduced aggregation propensity, fewer potential PTM motifs, and lower predicted immunogenicity using computational methods. Subsequently, we constructed these variants to analyze their expression levels and antigen-binding activities. The Ab417 variant H3L7 exhibited a higher expression level compared to Ab417.

Therefore, we further changed a residue in the LCDR3 of H3V7 to generate an even more improved variant (Ab612) with higher productivity and antigen-binding affinity. The Ab612 showed 2.6-fold higher productivity and improved biophysical properties, such as 1.4-fold increased purification yield, greater stability, lower aggregation propensity, 2-fold higher affinity for human L1CAM, and enhanced in vivo anti-tumor efficacy. Moreover, for the preclinical development of Ab612, we successfully generated a highly productive and stable research cell bank (RCB) and confirmed the scalability of the production process to a pilot scale.

Overall, the present results demonstrate that we successfully improved the biophysical properties and affinity of Ab417, generating an optimized antibody (Ab612). Ab612 is considered a promising candidate antibody for preclinical development.

## 4. Materials and Methods

### 4.1. Design of Ab417 Variants with Improved Biophysical Properties

#### 4.1.1. Sequence Analysis for Potential PTMs

Multiple alignments of Ab4 and Ab417 sequences to human germline sequences were generated using MAFFT [34], and entries in each alignment were ordered according to the sequence identity to the parental sequence. The antibody sequences were analyzed for potential PTMs such as Asn deamidation, Asp isomerization, N- and O-glycosylation, and oxidation.

#### 4.1.2. Construction and Comparison of 3D Models of Ab417 and Its Variants

Structural models of the Fv-region for Ab417, and variants thereof, were generated using Lonza Biologics’ modeling platform. Candidate structural template fragments for the FR and CDRs, as well as the full Fv, were scored, ranked, and selected from the antibody database based on their sequence identity to the target, as well as qualitative crystallographic measures (Å) of the template structure. In order to structurally align the CDRs to the FR templates, five residues on either side of the CDR were included in the CDR template. An alignment of the fragments was generated based on overlapping segments and a structural sequence alignment using MODELLER. An ensemble of structures that satisfy the conformational restraints derived from the set of aligned structural templates was created by simulated annealing and conjugation gradient optimization procedures. One or more model structures were selected from this ensemble based on an energy score derived from the quality of the protein structure and satisfaction of the conformational restraints. The models were inspected, and the side chains of the positions which differ between the target and template were optimized using a side chain optimization algorithm and energy minimized. A suite of visualization and computational tools were used to assess the conformational variability of the CDRs, as well as the core and local packing of the domains and regions and surface analysis to select one or more preferred models.

To assess the impact of different substitutions on affinity and stability, a number of structural criteria, including the solvent accessibility, local atomic packing, and location of the substitution relative to the predicted antigen-binding interface or the Fv dimer interface, electrostatic effects, and hydrogen bonding patterns, were used.

#### 4.1.3. Calculation of Aggregation Propensity and Assessment of Potential Substitutions

Aggregation hotspots were identified based on the sequence and structure of antibodies using Lonza’s AggreSolve^TM^ in silico tools. The intrinsic aggregation propensity score, A_res_, was calculated for overlapping 7-mer peptides, and the score was calculated over an entire amino acid sequence to generate A_tot_. In addition, S_res_, which reflects the aggregation propensity of a 7-mer peptide from its folded state, was calculated by applying the conformational correction to the intrinsic aggregation propensity profile [35]. A summary score, S_tot_, was calculated based on the position-specific S_res_. In addition, given that non-specific protein-protein interactions can be caused by aggregation-prone hotspots on the protein’s surface, the surface aggregation propensity per position, Tres, was calculated. A summary score, T_tot_, was calculated based on the position-specific Tres descriptors.

All positions outside the CDRs that were part of the hot spots were assessed based on their potential impact on binding affinity and stability. Each position was classified as either Neutral, Critical, or Contributing. A neutral position means that substituting another amino acid at this position should not affect binding affinity or stability negatively. A contributing position means that a substitution can be made, but the position may contribute to binding affinity or stability. A critical position means that the position risks a decreased binding affinity or stability, and therefore parental amino acid must be retained.

#### 4.1.4. Analysis of Th Epitopes

The epitopes or clusters of adjoining epitopes of Ab417 and engineered variants were analyzed using Epibase^TM^ for substitutions that would remove or reduce binding to HLA allotypes to the greatest extent possible, with a focus on the HLA-DR allotypes, because these are known to express at a higher level than the other allotypes DQ and DP [36]. Human germline sequences were not considered to be immunogenic as they are found in the pool of circulating antibodies. The substitutions in the engineered antibodies were analyzed for all chosen HLA allotypes (DRB1, DR3/4/5, DQ, and DP).

### 4.2. Construction of Expression Plasmids and Transient Expression of Antibodies

The genes coding for the VH or VL of the variants were synthesized and subcloned into the EcoRI-ApaI or HindIII-BsiWI sites of the expression plasmid (pCMV-dhfrC) containing the human Cγ1 or Cκ gene to construct heavy or light chain expression plasmid, respectively, as described previously [31].

For transient expression of antibodies, HEK293F cells (Gibco, Thermo Fisher Scientific, Waltham, MA, USA) or ExpiCHO^TM^ cells (Thermo Fisher Scientific, Waltham, MA, USA) were grown in FreeStyle 293 Expression medium (Gibco, Thermo Fisher Scientific, Waltham, MA, USA) at 8% CO_2_, 37 °C, 125 rpm or in ExpiCHO Expression medium (Thermo Fisher Scientific, Waltham, MA, USA) at 5% CO_2_, 37 °C, 110 rpm on a shaker. For transfection, the heavy and light chain expression plasmids (two vector system) or expression plasmid containing heavy and light chain (one vector system) were introduced into HEK293F using polyethyleneimine (Polysciences) or ExpiCHO™ cells (Thermo Fisher Scientific, Waltham, MA, USA) using the ExpiCHO Expression System. On day 6 post-transfection, the cell culture supernatant was centrifuged, filtered using a bottle top filter (0.22 µm PES, Sartorius, Germany), and subjected to quantitative ELISA and indirect ELISA to analyze the expression level and antigen binding activity.

### 4.3. ELISAs

A quantitative ELISA was performed to determine the concentration of antibodies. Anti-human IgG kappa antibody (Invitrogen, Waltham, MA, USA) was coated on the 96-well immunoplate (Thermo Fisher Scientific, Waltham, MA, USA) at 4 °C overnight. After blocking with 2% skim milk in 0.05% PBST (PBS with 0.05% tween 20), a standard IgG sample (400 ng/mL) or cell culture supernatant serially diluted in 0.1% PBA was added to each well and incubated at 37 °C for 1 h. The bound antibody was detected using anti-human IgG Fc-HRP (1:10,000 (*v*/*v*); Invitrogen, Waltham, MA), as described previously [31].

An indirect ELISA was performed to determine the antigen binding activity of the antibody. Purified human recombinant L1CAM (hL1-s1) and mouse recombinant L1CAM (mL1-s1) were prepared as described previously [31]. hL1-s1 or mL1-s1 (100 ng/well) was coated on each well at 4 °C overnight. Serially diluted cell culture supernatant or antibody was incubated with the hL1-s1 or mL1-S1 at 37 °C for 1 h. The bound antibody was detected using anti-human IgG(Fc-specific)-HRP (1:10,000, Invitrogen, Waltham, MA, USA).

A competitive ELISA was performed to determine the antigen-binding affinity of the antibody. hL1-s1 or mL1-s1 (100 ng/well) were coated on each well at 4 °C for 12 h. Ab417 or Ab612 antibody (10 ng/mL) in 0.1% PBA solution was pre-incubated with various concentrations (10^−12^~10^−7^ M) of hL1-s1 or mL1-s1 as a competing antigen at 37 °C for 3 h. The reaction mixture was added to each well coated with the hL1-s1 or mL1-s1, and indirect ELISA was carried out. Affinity (KD) was defined as the antigen concentration required to inhibit 50% of the antigen-binding activity.

A sandwich ELISA was performed to compare the antigen-binding activities of the Ab612 antibody samples produced from the transient expression and the RCB culture. Anti-human Fc antibody (Invitrogen, Waltham, MA, USA, 100 ng/well) was coated on each well at 4 °C overnight. Ab417 or Ab612 (50 ng/mL) was added to each well, incubated at 37 °C for 1 h, and further incubated with hL1-s1 serially diluted from 4 μg/mL at 37 °C for 1 h. The hL1-s1 captured by Ab417 or Ab612 was incubated with mouse anti-s1 antibody KR127 [37] at 37 °C for 1 h. The bound KR127 antibody was detected using anti-mouse IgG(Fc-specific)-HRP (1:10,000, Invitrogen, Waltham, MA, USA).

### 4.4. Purification of Antibodies

Ab417 and Ab612 were purified using a Protein A affinity column. Harvested Cell Culture Fluid (HCCF) containing a monoclonal antibody was loaded on to pre-packed protein A column (HiTrap MabSelect^TM^ Sure, Cytiva, Marlborough, MA, USA) equilibrated with binding buffer (20 mM sodium citrate, pH 6.0). The column was then re-equilibrated with binding buffer followed by a wash step at pH 6.0 and finally elution with sodium citrate buffer at pH 6.0 and 2.5 (gradient elution). The gradient elution experiments were carried out using a 30 column volume (CV) linear gradient from 0% to 100% buffer B (20 mM sodium citrate, pH 6.0). For virus inactivation, the elution fraction was incubated at 4 °C for 1 h and then neutralized with 1 M Tris-HCl (pH 9.0) to minimize the effect on the structure of the antibody under low pH conditions. The column was then regenerated and sanitized using 0.5 N NaOH.

Desalting chromatography (HiPrep™ 26/10 Desalting, Cytiva, Marlborough, MA, USA) was performed to replace the buffer in the primary purified product (affinity eluate) suitable for the loading condition (50 mM Na-Acetate, pH 5.0). Further purification of Ab417 and Ab612 was performed using cation-exchange chromatography. The sample was applied to a HiTrap Capto SP column (Cytiva, Marlborough, MA, USA) and eluted with a linear gradient of 0 to 500 mM NaCl, 30 CV. 

Pooled fractions were further concentrated using Amicon Ultra-4 Centrifugal Filter Unit with Ultracel-10 membrane (Merck Millipore, Burlington, MA, USA). A concentrated protein sample was loaded onto the HiLoad Superdex 200 pg column (Cytiva, Marlborough, MA, USA) equilibrated with Phosphate-buffered saline (PBS) buffer at the rate of 1.0 mL/min. The elution profile was analyzed by the absorbance at 280 nm.

The purity and aggregation of purified protein were determined using high-performance size exclusion chromatography (SE-HPLC). A Waters HPLC (Alliance 2695) system was used with a Bio SEC-3 column (3 μm, 300 Å, 4.6 × 300 mm, Agilent, Santa Clara, CA, USA) at 0.3 mL/min flow rate (isocratic) using a mobile phase buffer of 20 mM Na-phosphate w/150 mM NaCl, pH 6.8. 

### 4.5. Dynamic Light Scattering (DLS)

DLS was performed using Zetasizer (Malvern, Herrenberg, Germany). After the Z-average of antibody sample was measured at 25 °C, 50 μL of the sample was added to disposable cuvettes (ZEN0040, Malvern, Herrenberg, Germany) and gradually the temperature was increased from 30 °C to 80 °C with 3 °C of temperature interval and a fixed angle of θ = 173°. The Z-average and intensity were calculated using Zetasizer Software version 7.02 (Malvern, Herrenberg, Germany).

### 4.6. Affinity Determination of Antibodies Using Octet red384 System

The affinity of the antibody was determined using the Octet Red384 system (Sartorius, Goettingen, Germany). Anti-human IgG sensor AHC (ForteBio, Fremont, CA, USA) was firstly soaked in 0.1% PBA for 20 min. The antibody (0.2 mL of 0.5 µg/mL) was captured for 10 min followed by washing with 0.1% PBA for 2 min. hL1-s1 or mL-s1 (25, 12.5, 6.25, 3.125, or 1.5625 nM in 0.1% PBA) was then incubated with the antibody captured on the sensor. Association and dissociation rates were measured for 10 min and 30 min, respectively. For correction of baseline drift, a control sensor was designated as an antibody-captured AHC sensor exposed to running buffer only. All analytes were recalculated by subtraction of the rate of a control sensor. The operating temperature was maintained at 30 °C and agitated at 1000 rpm. Data were analyzed using a 1:1 interaction model (fitting global, R_max_ unlinked by a sensor) with analysis software (ForteBio, ver. 8. 2).

### 4.7. Capillary Isoelectric Focusing (cIEF) Analysis

cIEF was conducted according to the SCIEX application protocol. All cIEF experiments were performed on SCIEX PA800 plus instrument with a 50 µm i.d. neutral coated capillary (SCIEX P/N 477441) at the length of 30.2 cm. The UV detector was used to detect absorbance at 280 nm wavelength. A cIEF master mix solution was composed of 3 M urea-cIEF gel solution, pharmalyte 3–10 carrier ampholytes, cathodic stabilizer (500 mM arginine), anodic stabilizer (200 mM iminodiacetic acid), and five pI markers (10.0, 9.5, 7.0, 5.5, 4.1). Analytes (5 mg/mL) were also mixed with 10 µL of the master mix solution. All experiments were performed in triplicate. The pI values of the sample were calculated using qualitative analysis of a 32 Karat software.

### 4.8. In Vivo Antitumor Activities of Ab417 or Ab612

All the animals were housed under a 12/12 h light/dark cycle (light phase, 8:00 A.M. to 8:00 P.M.) with a standard laboratory diet and water ad libitum. All animal handling and experiments were conducted with the approval of the Institutional Animal Care and Use Committee (IACUC) of preclinical CRO Biotoxtech (180678). Nude mice (BALB/cSlc-nu, 5 weeks old) were obtained from Japan SLC, Inc (Shizuoka, Japan). Choi-CK cells (1 × 10^6^) were inoculated into the right flank of each mouse. Constructed Choi-CK tumor tissue (3 × 3 × 3 mm^3^) was subcutaneously inoculated into the back of mice. After tumor volume reached a mean of 100 mm^3^ (*n* = 8 per group), the antibody at a dose of 10 mg/kg was i.v. injected 3 times per week for 3 weeks. Tumor growth was monitored by measuring the length and width of the tumor with a caliper and calculating tumor volume based on the following formula; TV (mm^3^) = L (mm) × W2 (mm^2^) × 1/2, where L is length and W is width. Body weight was measured twice a week, and tumor tissues were taken out and weighed at the end of the experiment. Tumor growth inhibition rate (IR) was calculated as the following formula; IR (%) = (1 − T/C) × 100. T is the mean tumor weight of the antibody treated group, and C is the mean tumor weight of the mock control group.

Data were validated using SAS (Version 9.3, SAS Institute Inc., Cary, NC, USA). Each point represents the mean ± SD. Statistical comparison between groups were performed by one-way analysis of variance (ANOVA) followed by Dunnett’s t-test.

### 4.9. Production of Ab612 from Research Cell Bank (RCB)

#### 4.9.1. Construction of RCB

RCB was generated by Shanghai OPM Biosciences Co., Ltd. (Shanghai, China). CHO-K1 cells were grown in CD CHO Medium (GIBCO, Thermo Fisher Scientific, Waltham, MA, USA) containing 6 mM L-glutamine (Sigma, St. Louis, MO, USA). Cells were incubated under the condition of 120 rpm, 37 °C, and 8% CO_2_. Freshly prepared linearized plasmid was transfected into CHO-K1 cells by electroporation using a Bio-Rad system. For each sample, 1 × 10^7^ cells were transfected with a total of 40 μg of the linear plasmid. At 24 h later, the transfected cells were plated into a 96-well plate with 4000 cells/well. The culture medium was CD CHO Medium containing 50 μM L-Methionine sulfoximine (MSX) (Sigma, St. Louis, MO, USA) and 1 × GS-Supplement (Sigma, St. Louis, MO, USA). Cells were statically cultured in an incubator with 8% CO_2_ and 37 °C. The plasmid was transfected into cells twice by electroporation, and then the cells were plated into 20 pieces of a 96-well plate. Mini pool fed-batch assay medium was OPM-CHO CD07 Medium (OPM, China) with 1:200 anti-clumping Agent (ACA, OPM, China). The CDF18 (OPM, China) at 3%, 5%, 6%, 6% and 5% concentration and CDF26 (OPM, China) at 0.3%, 0.5%, 0.6%, 0.6% and 0.5% were fed on day 3, 5, 7, 9, and 11, respectively. The glucose was fed as needed to maintain at 2–6 g/L. The fed-batch was stopped when the viability is about 60% or on day 12.

#### 4.9.2. Stability Study of Research Cell Bank (RCB)

In suspension culture from RCB stock, cells were passaged once every 3 days for 69 days with 0.4 × 10^6^ cells/mL seeding density in OPM-CHO CD07 Medium with 50 μM MSX and 1:200 Anti-Clumping Agent. In a fed-batch assay, cells were incubated at the density of 0.8 ± 0.1 × 10^6^ cells/mL with 30 mL culture volume. The medium was OPM-CHO CD07 Medium with 1:200 ACA. The CDF18 was fed on day 3, 5, 7, 9 and 11 at 3%, 5%, 6%, 6% and 5%, respectively, and CDF26 was fed at 0.3%, 0.5%, 0.6%, 0.6% and 0.5%. The glucose was fed as needed to maintain it at 2–8 g/L. The fed-batch was stopped when the viability is about 60% or on 12 days. The culture supernatants were harvested by centrifuging at 3500 rpm for 30 min under 4 °C.

#### 4.9.3. Production and Purification of Antibodies

The HCCF obtained from RCB production was purified using protein A (Repligen, Waltham, MA, USA) packed in XK26/20 column (Cytiva, Marlborough, MA, USA) affinity chromatography. The purified antibody was stored in 10 mM sodium phosphate buffer containing 5% sorbitol and 0.01% tween 20 and dialyzed using HiPrep^TM^ 26/10 Desalting Column (Cytiva, Marlborough, MA, USA). The concentration of purified antibody was determined with a Nanodrop (Thermo fisher scientific, Nanodrop 2000) based on the molar extinction coefficient. The produced Ab612 from RCB was used for analyzing purification by SEC-HPLC and antigen-binding activity by ELISA.

## 5. Conclusions

In conclusion, we successfully improved the biophysical properties and affinity of Ab417 through antibody engineering based on computational methods, which generated an optimized antibody, Ab612. This antibody exhibited enhanced in vivo anti-tumor efficacy compared to Ab417. Additionally, we successfully generated a highly productive and stable RCB and confirmed the scalability of the production process to a pilot scale. Ab612 is considered a promising candidate antibody for preclinical development.

## Figures and Tables

**Figure 1 ijms-22-06696-f001:**
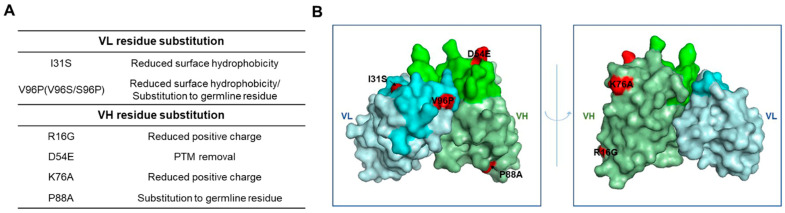
Schematic representation of the substitutions of Ab417 for improved biophysical properties. (**A**) Design of the VL and VH residue substitutions. (**B**) The 3D model of Ab417. The I31S and V96P in the VL and the R16G, D54E, K76A, and P88A in the VH are shown in red. Gray, VL; Green, VH; Lime, LCDR; Light green, HCDR.

**Figure 2 ijms-22-06696-f002:**
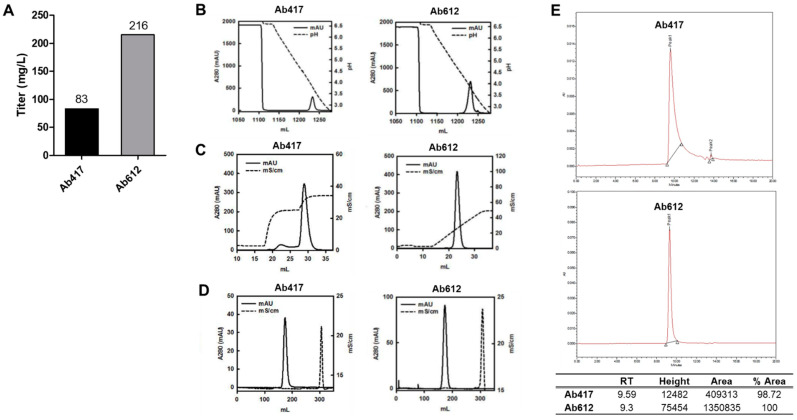
(**A**) Expression levels of antibodies using the ExpiCHO transient expression system. (**B**,**C**) Ab417 and Ab612 were purified by protein A affinity chromatography (**B**) and CIEX by stepwise gradient elution and linear gradient elution, respectively (**C**). (**D**) Final polishing was conducted using SEC. (**E**) SEC-HPLC of each antibody was performed to determine the purity and homogeneity following purification.

**Figure 3 ijms-22-06696-f003:**
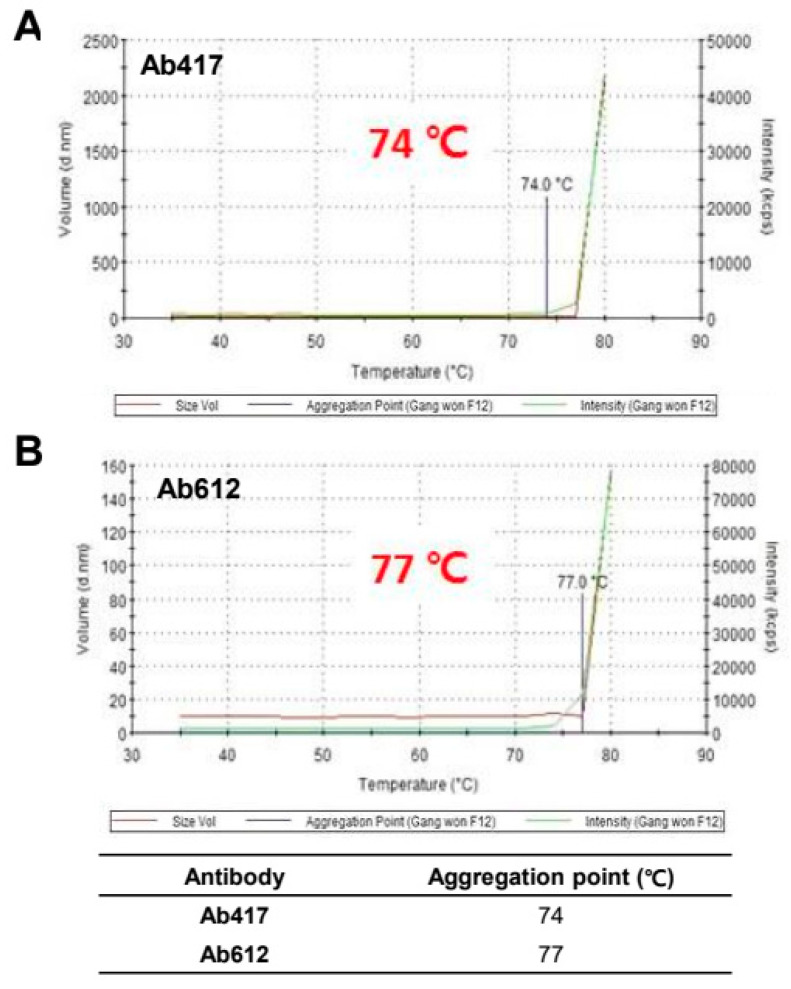
Assessment of thermal stability of Ab417 (**A**) and Ab612 (**B**) by DLS. The blue line indicates the starting point of protein aggregation.

**Figure 4 ijms-22-06696-f004:**
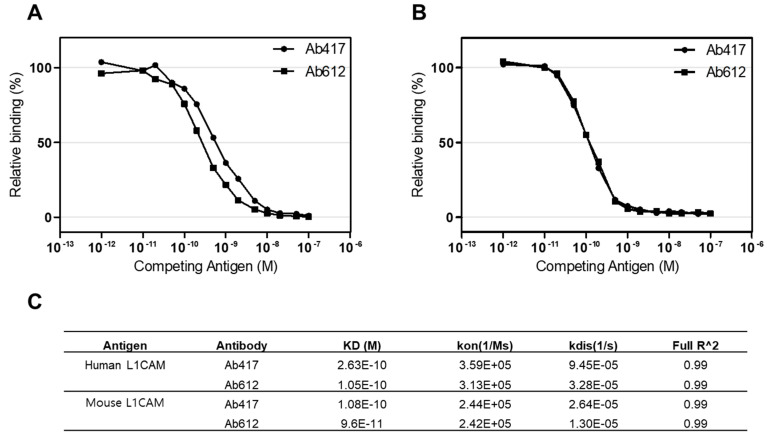
Determination of the affinities of Ab417 and Ab612 for hL1-s1 (**A**) and mL1-s1 (**B**) by competitive ELISA and the Octet system (**C**). Kon, rate of association; Kdis, rate of dissociation; Full R^2, estimate of the goodness of the curve fit.

**Figure 5 ijms-22-06696-f005:**
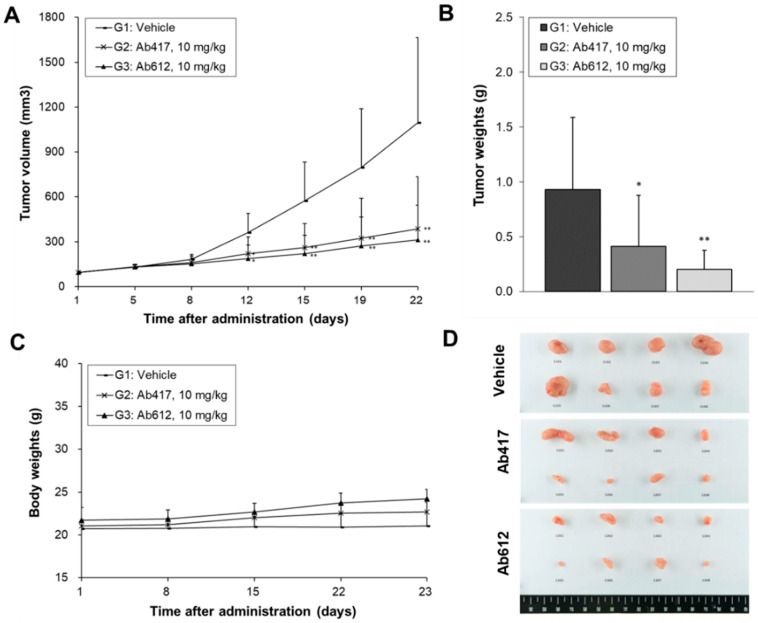
Anti-tumor efficacy of Ab612 compared with Ab417 in a Choi-Ck cholangiocarcinoma xenograft nude mouse model (*n* = 8). When the tumor volume reached an average of 100 mm^3^, dosing (10 mpk) was initiated 3 times weekly for 22 days. Mean tumor volume (**A**), tumor weight (**B**), and body weight (**C**) are shown. (**D**) Photographs of the resected tumors at the end of the experiment. Each point represents the mean ± SD. * *p* < 0.05, ** *p* < 0.01, significant difference from the isotype control group by Dunnett’s t-test.

**Figure 6 ijms-22-06696-f006:**
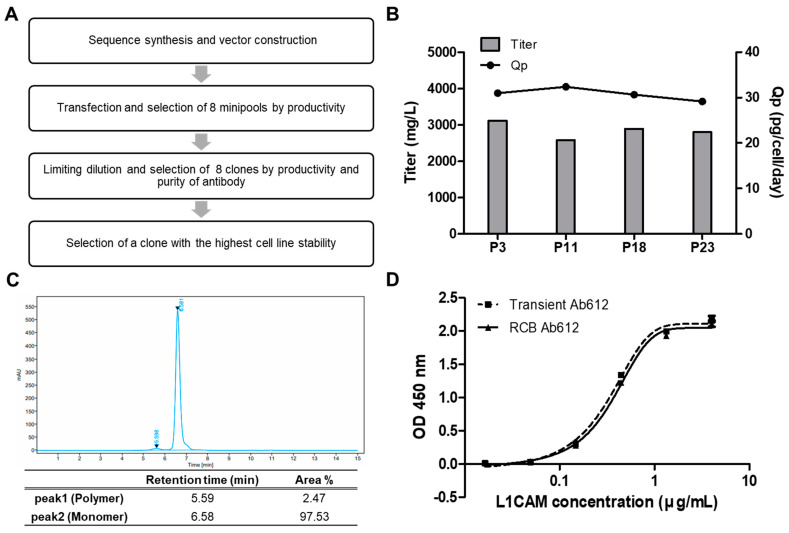
(**A**) Schematic diagram showing the generation of a research cell bank (RCB). (**B**) Ab612 titer was measured to assess the expression level, and Qp was calculated as PCD = (pg/cell/day) using PA-HPLC. (**C**) The purity of Ab612 produced from the RCB was analyzed by SEC-HPLC after purification. (**D**) Ag binding activity was determined by sandwich ELISA.

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
