# Peer review of "Improvement of Biophysical Properties and Affinity of a Human Anti-L1CAM Therapeutic Antibody through Antibody Engineering Based on Computational Methods"

_ijms, 2021, doi:10.3390/ijms22136696_

Round 1

Reviewer 1 Report

Review of  Manuscript: Improvement of Biophysical Properties and Affinity of a Hu-2 man Anti-L1CAM Therapeutic Antibody Through Antibody 3 Engineering Based on Computational Methods by Chae et al.

This manuscript is concerned with improving the yield and affinity of a basic anti-L1 adhesion surface protein antibody (Ab 417) used to kill and/or block growth of tumor cells (in this case, cholangiocarcinoma). The authors have used theoretical approaches to identify sites of possible post-translational modification, surface charge reduction, sites that may be active in inducing cell-mediated immunity and sites most likely to cause protein aggregation. The authors present experimental data that support their approach in that they have engineered a form of the antibody (Ab 612) that has a higher affinity for human  L1 (by a factor of 2) and which is expressed in expression cells significantly faster than the original antibody. Use of light scattering indicates that there is significantly less aggregation of Ab 612 than of Ab 417. The authors have presented their approach and results in a clear and logical manner. I recommend publication.

They may wish to answer the following questions:

1.Do the two antibodies kill the cancer cells or do they block cell division?

2.Do these antibodies kill and/or block division of normal cells?

3.Is the factor of 2 for the difference in affinities for human L1 statistically significant?

Author Response

1.Do the two antibodies kill the cancer cells or do they block cell division?

  The antibodies inhibited tumor cell proliferation in vivo, but not in vitro.

2.Do these antibodies kill and/or block division of normal cells?

  The antibodies did not inhibit normal cell proliferation in vitro. We think that the antibodies do not inhibit normal cell proliferation in vivo because they did not induce any adverse event in rats and monkeys in toxicity studies.

3. Is the factor of 2 for the difference in affinities for human L1 statistically significant? We think that it is statistically significant because we measured their affinities twice and observed that SD of KD value (0.26 or 0.10 nM) of Ab417 or Ab612 was ± 0.014.

Reviewer 2 Report

The authors present a nice example of a combined computational/experimental approach to improve the properties of a human anti-L1 CAM mAb. The methods are well exposed, the discussion, well presented and the conclusions are very promising. 

A nice example of design and development.

Author Response

A nice example of design and development.

  We express our deepest gratitude for good comment on our work.